# Reasons for Encounters and Comorbidities in Adolescents with Intellectual Disability in General Practice: A Retrospective Analysis of Data from the Ask Study

**DOI:** 10.3390/children10091450

**Published:** 2023-08-25

**Authors:** Menghuan Song, Tran T. A. Le, Simon Denny, Nicholas G. Lennox, Lyn McPherson, Robert S. Ware, David Harley

**Affiliations:** 1Queensland Centre for Intellectual and Developmental Disability, Mater Research Institute-University of Queensland, Brisbane, QLD 4101, Australia; menghuan.song@uq.net.au (M.S.); or lttanh@huemed-univ.edu.vn (T.T.A.L.); n.lennox@uq.edu.au (N.G.L.); d.harley@uq.edu.au (D.H.); 2Institute of Chinese Medical Sciences, University of Macau, Taipa, Macau SAR, China; 3Department of Psychiatry, University of Medicine and Pharmacy, Hue University, Hue 530000, Vietnam; 4Mater Young Adult Health Centre, Mater Hospitals, Brisbane, QLD 4072, Australia; simon.denny@mater.org.au; 5Menzies Health Institute Queensland, School of Medicine and Dentistry, Griffith University, Brisbane, QLD 4111, Australia; lyn.mcpherson@griffith.edu.au; 6Centre for Clinical Research, University of Queensland, Brisbane, QLD 4006, Australia

**Keywords:** adolescent, chronic disease, incidence, intellectual disability, prevalence, primary health care

## Abstract

Adolescents with intellectual disability have substantial health needs. This retrospective analysis of data from the Ask Study describes reasons for primary care encounters and the prevalence and incidence of chronic physical and mental conditions among a cohort of community-dwelling adolescents with intellectual disability. Participants attended secondary schools in southern Queensland, Australia. Primary care data were extracted from primary care records. Demographic and health information was collected using carer-completed questionnaires. Reasons for primary care encounters, disease prevalence at age 16 years, and disease incidence through adolescence were reported. Data were obtained for 432 adolescents with intellectual disability (median follow-up: 4.1 years). Skin problems (29.4 per 100 encounters) were the most common reason patients presented for primary care, followed by psychological and behavioural problems (14.4 per 100 encounters) and musculoskeletal problems (13.8 per 100 encounters). Conditions with the highest prevalence were autism spectrum disorder (18.6%) and asthma (18.1%). The prevalence of epilepsy, visual impairment, and cerebral palsy were 14.7, 11.1, and 8.0%, respectively. Gastroesophageal reflux had the highest incidence (9.4 cases per 1000 person-years). Adolescents with intellectual disability have significant healthcare needs, which general practitioners need to be aware of and address. Study findings should inform the development of training programs for general practitioners.

## 1. Introduction

People with intellectual disability continue to experience substantial health disparities and healthcare barriers, despite significant improvements in healthcare services for this marginalized population in recent decades [1]. Adolescents with intellectual disability have significant health demands, with higher rates of physical conditions and mental disorders than their non-disabled peers [2,3,4] and considerable unmet health needs [5]. The transition from adolescence to adulthood, including the transition from the paediatric to the adult health care system, is particularly challenging [5,6,7,8]. Not only are there biological changes around puberty, but also specific health and developmental demands [9], which may be complicated due to intellectual disability. For example, mental disorders may present but can be very difficult to diagnose due to communication difficulties. The healthcare system generally does not adequately meet the needs of adolescents with intellectual disability [5], with consequent poorer health status in adulthood [10]. Improving the quality of primary care is crucial to reducing unmet health needs and facilitating better health outcomes for this population. Data on problems managed during general practitioner (GP) encounters can reflect individual demands and thus provide insights into optimizing general practice.

In Australia, the first point of contact in the health care system is usually a GP at the primary care level. The cost of a GP visit is covered wholly or partly by Medicare, a federally funded universal medical insurance scheme. While the scheme does allow for longer consultations for patients with intellectual disability, this is limited to one annual review. It is challenging for GPs to assess patients with intellectual disability during routine encounters due to their complexity, possible communication difficulties, and the high prevalence of co-morbidities. 

In the general population of adolescents, general, respiratory, musculoskeletal, skin, and mental problems are common in GP encounters [11,12,13]. In people with intellectual disability of all ages, the most common presenting problems in primary care encounters are mental problems, general, respiratory, endocrine, and metabolic problems [14,15]. However, only limited data describing common presenting problems of adolescents with intellectual disability in primary care are available. 

Chronic conditions are the main causes of poor health, disability, and death [16]. They often require a long period of supervision, observation, or care, and they account for the bulk of healthcare expenditure [16,17]. Primary care is key in preventing and treating these conditions [18,19]. Profiles of chronic conditions in the general population of Australian adolescents are well-reported. Anxiety disorder (prevalence range: 6.3–20.4%) [20,21], depressive disorder (4.3–14.0%) [20,21], and asthma (10.7%) [20] are common in Australian adolescents. However, there is limited information on the profile of chronic conditions of adolescents with intellectual disability. This information is needed to help GPs identify, manage, and promote optimal health and well-being for this patient cohort, especially considering the often inadequate training for GPs in caring for people with intellectual disability [22]. 

The aim of this study is to describe problems managed during primary care encounters and the prevalence and incidence of chronic medical conditions in Australian adolescents with intellectual disability. 

## 2. Materials and Methods

### 2.1. Study Population

This prospective cohort study of adolescents with intellectual disability was conducted in southern Queensland, Australia, between 2006 and 2010. Data were originally collected as part of the Ask Study, a randomized controlled trial investigating whether a health intervention package could improve health outcomes and self-determination in health (ClinicalTrials.gov Identifier: NCT00519311) [23]. The intervention did not influence disease diagnosis or GP attendance; consequently, primary-care-level data from the intervention and control groups were combined for this analysis. 

Eligible participants were secondary school students aged 10–20 years as of 1 January 2006 and attended either separate Special Education Schools (SES) or Special Education Units (SEU). Students could attend these schools if they had been assessed by Education Queensland as having an intellectual disability. Students with less severe disability attended SEUs, which were located within conventional secondary school campuses, while adolescents with more severe intellectual disability or multiple disabilities attended SESs. 

Informed consent was obtained from all principals or heads of schools, carers, and GPs involved in this research. The study was conducted according to the guidelines of the Declaration of Helsinki. Ethics approval was granted on 14 December 2005 by the Queensland Government Department of Education and the Arts (File No: 550/27/424) and the University of Queensland Behavioural and Social Sciences Ethical Review Committee (Clearance No: 2004000081).

### 2.2. Data Collection

Demographic information, school types, and clinical characteristics of participants were obtained from mailed questionnaires completed by carers in May 2007. Clinical characteristics collected included cause of intellectual disability and carer ranking of general health and physical mobility. Data from primary care consultations were collected from GP notes by research staff between 10 September 2009 and 10 June 2010. These data included patient and family history, hospitalizations, diagnoses, symptoms, and case-finding activities. These data were available from 1 January 2006 to the data collection date. Patient histories, diagnoses, and symptoms were extracted by an experienced paediatric nurse, and new diagnoses (incident cases) were identified. These were reviewed by a GP who had extensive experience in intellectual disability medicine. Information on upper respiratory tract infections and some unspecified clinical tasks, such as when the GP was required to confirm a diagnosis for administrative purposes, were not extracted from primary care records because these were unnecessary for the Ask Study aims.

### 2.3. Measurement of Outcomes

Patient histories, diagnoses, and symptoms were coded using both the International Classification of Primary Care, Second Edition Revised (ICPC-2R) systems and the International Statistical Classification of Diseases and Related Health Conditions, 10th Revision (ICD-10) by authors T.T.A.L. and L.M. [24,25]. Reasons for primary care encounters include diagnosis or management of acute and chronic conditions and other activities such as contraception and suspected pregnancy management. The ICPC-2R system was used to classify encounters for all participants throughout the study period. Symptoms were coded only if no diagnosis was obtained during the GP encounter. 

Definite diagnoses of chronic physical conditions or mental disorders were classified using the ICD-10 system. Incident cases of chronic physical or mental disorders were determined to be newly identified if they had not been noted in previous clinical records. In particular, “active epilepsy” in this paper was defined as epilepsy requiring medical or surgical management during the follow-up period. “Visual disturbance or impairment” was defined as blindness or visual impairment uncorrected by glasses.

### 2.4. Statistical Analysis

Descriptive statistics were calculated for characteristics and medical conditions of participants. Problems in primary care encounters were reported as rates per 100 GP encounters. Prevalence of chronic physical conditions or mental disorders was calculated for participants whose GP data were available within one year of their 16th birthday. Age of 16 years was chosen because puberty is usually complete [26]. A generalized linear model with a binomial family and an identity link, and robust standard errors (clustered on individual GP) was used to calculate the associations between gender and the prevalence of reported chronic physical conditions and mental disorders, with effect estimates presented as risk difference (95% confidence interval; CI). Incidence was calculated for each condition or health state. These were calculated using Poisson regression with models offset by the natural logarithm of the time-at-risk and were presented as cases per 1000 person-years. Analyses were conducted using Stata v16 (StataCorp LLC, College Station, TX, USA).

## 3. Results

Primary care records were collected for 432 participants, 415 of whom had records within one year of their 16th birthday. The median follow-up time was 4.1 years, with 429 participants followed for at least 3.2 years. The total follow-up time was 1776.3 person-years. Demographic characteristics are presented in Table 1; 46.1% of the 432 participants were female, and 63.2% were aged 14 to 16 years in 2006.

### 3.1. Reasons for Primary Care Encounters

Information on 3670 GP encounters was coded from primary care records during the study period. Participants had a mean of 2.1 GP encounters per year. The most common reason for encounters was skin problems (29.4 per 100 GP encounters). Skin problems consisted of infected digits (3.9%), cellulitis (2.8%), warts (2.2%), and acne (2.2%). Other common reasons for encounters were mental and behavioural problems (14.4 per 100 GP encounters) and musculoskeletal problems (13.8 per 100 GP encounters) (Table 2). 

### 3.2. Prevalence of Chronic Physical Conditions or Mental Disorders

Among the 415 participants whose clinical notes were available at age of 16 years, 347 (83.6%) had at least one chronic physical condition or mental disorder, with 40.7% and 69.9% having at least one mental or physical condition, respectively (Table 3). Among all the reported specific mental disorders, autism spectrum disorder (ASD) had the highest prevalence (18.6%), followed by attention deficit hyperactivity disorder (ADHD: 14.7%). The prevalence of ASD and ADHD was higher in male than female participants (26.0% and 18.4% in males vs. 9.9% and 10.4% in females, respectively). Anxiety and depressive disorders were noted in 3.9% and 3.6% of adolescents. 

The prevalence of neurological, respiratory, endocrine and metabolic, visual, musculoskeletal, skin, and heart conditions at age 16 were 28.7, 19.0, 15.2, 13.5, 13.0, 9.4, and 9.2%, respectively. There were 22.9% of adolescents with congenital malformations and 7.5% with cardiac defects. Prevalence of asthma, epilepsy, visual impairment, cerebral palsy, and gastroesophageal reflux were 18.1, 14.7, 11.1, 8.0, and 4.3%, respectively.

### 3.3. Incidence of Medical Conditions

Among chronic medical conditions, gastroesophageal reflux disease had the highest incidence (9.4 cases per 1000 person-years), followed by atopic dermatitis (8.8) and depressive disorder (7.7) (Table 4). Incidences of hearing loss, epilepsy, and visual impairment were 4.8, 3.5, and 2.5 cases per 1000 person-years, respectively. 

## 4. Discussion

### 4.1. Principal Findings

Adolescents with intellectual disability have significant health needs, especially involving the management of skin, mental and behavioural, and musculoskeletal problems. A high prevalence of chronic diseases was reported, with mental disorders in about two-fifths and physical conditions in more than two-thirds of adolescents. About one-fifth and one-seventh had ASD and ADHD recorded in their primary care records. Males were more likely than females to have ASD and ADHD. Anxiety or depressive disorders were noted in fewer than 5% of adolescents. Asthma (about one-fifth), epilepsy (one-seventh), visual impairment (one-tenth), and cerebral palsy (one-twelfth) were common physical conditions. Gastroesophageal reflux had the highest disease incidence.

### 4.2. Comparison with the Literature on General Population

#### 4.2.1. Primary Care Encounters

Mental problems were more common in primary care encounters for Ask Study participants (14.4% of the encounters) than for general Australian adolescents (1.9–8.1%) [12,13]. The differences are likely to be due to special needs in the management of behavioural problems in people with intellectual disability. In our study, behavioural problems were often the reason for medical encounters. This is not the case for the Australian population as a whole [12,13]. 

Skin problems (29.4% encounters) were the commonest reason for GP encounters among Ask Study participants, comparable to the rate in Booth’s study (17.4–28.2%) [13] but higher than the rate (10.4%) from Haller’s study [12]. Warts, acne, rash, and dermatitis are the main subcategories of skin problems in our and Booth’s studies [13]. Warts and rashes were not reported in Haller’s study [12]. In our study, cellulitis was the most common specific skin problem requiring primary care. 

#### 4.2.2. Prevalence of Chronic Conditions

Overall mental disorders are more prevalent; whereas anxiety and depressive disorders are less prevalent in our participants (40.7%; 3.9%, and 3.6%, respectively) than the estimates in general Australian adolescents aged 12–17 years (12.8–15.9%; 6.3–7.7% and 4.3–5.8%, respectively) [21], which is in line with Mazza’s study done in people with intellectual disability at all ages [27]. Mazza’s study suggested people with an intellectual disability rarely present with neurotic disorders [27]. Difficulties in diagnoses of psychiatric diseases in people with intellectual disability may contribute to these prevalence differences. Psychiatric diseases can be precipitated by physical conditions and psycho-social factors [28] and sometimes present as behavioural problems in people with intellectual disability [29]. Psychiatric diseases should only be diagnosed after a comprehensive health check and thorough environmental investigation. Behavioural problems were common reasons for GP encounters for our participants. Complex health conditions, communication barriers, and generally inadequate training of GPs make diagnosis time-consuming and difficult.

Among physical conditions: neurological, endocrine and metabolic, musculoskeletal, skin, and heart conditions are more common in our participants (28.7, 15.2, 13.0, 9.4, and 9.2%, respectively) than in general Australian adolescents aged 15–24 years (4.6–7.4, 2.3–2.9, 8.0–10.2, 3.4–4.8, and 1.6–3.3%, respectively) [20,30,31]. The prevalence of congenital malformation or deformation differs markedly (22.9% in our participants vs. 0.6–1.2% in adolescents aged 15–24 years) [20,30,31] (Table A1 in Appendix A). Epilepsy (14.7% in our participants vs. 0.4–0.7% in general adolescents), asthma (18.1% vs. 10.6–10.7%), hearing loss (4.8% vs. 1.2–3.7%) and gastroesophageal reflux (4.3% vs. 3.4%) were more common in our participants than the general population. [20,30,31,32]. These conditions have been found to be more common in people with intellectual disability at all ages compared with people without intellectual disability [33,34]. 

### 4.3. Comparison with the Other Literature on People with Intellectual Disability

#### 4.3.1. Primary Care Encounters

The most common reason for GP encounters was skin problems, followed by psychological and behavioural problems and musculoskeletal complaints. In contrast, Weise’s studies indicated that in people with intellectual disability of all ages, the most common presenting problems were psychological problems, general and unspecified, and respiratory problems [14,15]. This is perhaps because of hormonal changes and injuries that make acne and musculoskeletal problems common during adolescence [35,36,37]. Unlike the Ask Study, the research by Weise is not a prospective cohort study, but a retrospective study that attempts to identify people with intellectual disability through intellectual-disability-related reasons for GP encounters or problems managed by GPs during follow-up [14], and so is less likely to be population representative. 

#### 4.3.2. Prevalence of Chronic Conditions

The prevalence of chronic mental disorders (40.7%) in our study is comparable to the estimates from two systematic reviews and meta-analyses on children, adolescents, and adults with intellectual disability (33.6–49.0%) [27,38]. We report a higher prevalence of ASD (18.6%), and a lower prevalence of ADHD (14.7%) than the community-based study of Oeseburg (ASD: 10.9%, ADHD: 21.1%) [2], but our prevalence of ASD is similar to that found in a population-based study in Western Australia (22.4%) [39]. 

Our study reports a higher prevalence (69.9%) of chronic physical conditions than Oeseburg (21.7%) [2], which may be due to differences in study design. In Oeseburg’s study, medical conditions were reported by carers for one year, while our study is a cohort study with a median follow-up of more than four years, utilizing GP clinical notes and including all chronic physical medical conditions.

The prevalence of epilepsy in our study was 14.7%, compared with 5.3% and 36.7% from other studies of adolescents with intellectual disability [2,4]. The prevalence of asthma (18.1%) and cerebral palsy (8.0%) in our study is higher than Oeseburg (9.9% for asthma, chronic bronchitis, and chronic obstructive pulmonary disease combined; and 0.5% for cerebral palsy) [2]. This may be because adolescents with intellectual disability often have complex health problems, and carer records may be incomplete. We report a similar prevalence of visual disturbance or impairment (11.1%) but a lower prevalence of hearing loss (4.8%) compared with a community-based study (10.2–13.3% and 6.7–10.4%, respectively) using information from carers [3]. 

#### 4.3.3. Incidence of Chronic Conditions

The incidence of visual impairment, hearing impairment, epilepsy, and diabetes were 3.2, 2.6, 0.8, and 0.5 per 100 person-years in Van Schrojenstein Lantman–de Valk’s study [40]. We calculated these incidences of 0.3, 0.5, 0.4, and 0.0 per 100 person-years, respectively. The lower rates may be because we enrolled a community-based cohort of adolescents, while Van Schrojenstein Lantman–de Valk’s considered an institutionalized sample of people with intellectual disability at all ages [40]. 

### 4.4. Implications

This study fills a significant gap in the literature by describing reasons for primary care encounters, and the prevalence and incidence of chronic physical conditions, and mental disorders among Australian adolescents with intellectual disability. The findings may help with the optimization of general practice and healthcare systems to provide care for adolescents with intellectual disability. 

Adolescents with intellectual and/or other developmental disorders often have communication barriers and complex health profiles (including physical and psychiatric diseases and behavioural problems). GPs need to be aware of the high prevalence of these conditions and provide early detection and effective treatment. However, at present GPs usually have insufficient training and experience. GP education could be improved by targeting developmental disability training to common problems and conditions. There should be a particular focus on mental disorders. High-quality primary care is in demand for people with comorbid intellectual disability and ASD or ADHD. In clinical practice, GPs need appropriate training in the diagnosis of psychiatric diseases, determination of environmental, physical, and mental causes of behavioural problems, and appropriate management of psychiatric diseases and behavioural problems. 

The health needs of people with intellectual disability have been shown to differ substantially from those of the general population, suggesting that it is important to establish specialised units for this population. Currently, there are few within Australia, and they are not sustainably funded. Our study suggests that the needs of adolescents and adults with intellectual disability differ. Specialized units would provide appropriate care tailored to these needs. They would also be best placed to assist a young person’s transition from child to adult mainstream services. 

Annual health checks have been recommended for people with intellectual disability. Health checks are the only primary-care level intervention shown to significantly increase health promotion and disease prevention activities in this population [41,42]. Health checks have proven to be effective and valid for case detection and disease prevention in people with intellectual disability [23,43,44], but without leading to increased consultation or medication costs [45]. It has been widely used in primary care in Queensland and is broadening its coverage [46]. Our results would be useful for both informing the development of GP training curricula and refining health assessments to meet this population’s health needs. 

### 4.5. Strengths and Limitations

Our study describes the prevalence and incidence of chronic physical conditions and mental disorders and the association between gender and some developmental disorders, namely ASD and ADHD, among Australian adolescents with intellectual disability. This study also reports problems managed by GPs, uniquely among studies in adolescents with intellectual disability. Our dataset is large for this field, community-based, relies on primary care records, and was extracted by experts in this field; moreover, the follow-up was lengthy.

Adolescents who participated in this cohort had similar ages (mean (standard deviation) of 15.5 (1.6) years on 1 July 2007) and sex distribution (female: 46.1% participants) to all eligible high school adolescents (age: 15.4 (1.6) years, female: 38.8%) with intellectual disability in southern Queensland. We have previously shown drop-out in the Ask Study was not associated with participant characteristics [47]. Therefore, we believe our results are generalisable.

Prevalence and incidence may be underestimated as participants could receive medical attention from a doctor other than their regular GPs or may not receive any medical attention for some problems. Upper respiratory tract infections and some unspecified clinical tasks were not extracted, and these often make up a significant proportion of general practice encounters [48]. Consequently, the absolute rate of other problems managed by GPs will be inflated, but the relative frequencies among the health problems we studied are meaningful. Because upper respiratory tract infections are non-chronic in nature, this will not affect the results of the prevalence and incidence of chronic medical conditions and mental disorders reported. A limitation may be the age of the data, which was collected from 2006 to 2010; however, we believe these data remain relevant as the health conditions of adolescents with intellectual disability and their reasons for primary care encounters are unlikely to have changed significantly since these data were collected. An additional limitation is that we were not able to collect data on the degree of intellectual disability in individual participants. More research is required to examine the association between the level of disability and chronic physical conditions and mental disorders.

## 5. Conclusions

Adolescents with intellectual disability have significant health needs. GPs need to be aware of the high prevalence of some chronic conditions. Healthcare programs for early detection and effective treatment of this population are needed. These results should inform the design of the GP training curriculum to better meet this population’s health demands.

## Figures and Tables

**Table 1 children-10-01450-t001:** Participant characteristics (N = 432).

Characteristic	n (%)
Total	432 (100.0)
Female	199 (46.1)
Age (years) on 1 January 2006	
10–13	115 (26.6)
14–16	273 (63.2)
17–20	44 (10.2)
School type	
Special Education School	235 (54.4)
Special Education Unit	197 (45.6)
Cause of intellectual disability	
Down syndrome	66 (15.3)
Other known cause	235 (54.4)
Unknown cause	131 (30.3)
General health status ranked by carers	
Excellent	115 (26.6)
Very good	145 (33.5)
Good	129 (29.9)
Fair	37 (8.6)
Poor	6 (1.4)
Physical mobility	
Completely independent	370 (85.6)
Requires assistance (aids or carer)	62 (14.4)

**Table 2 children-10-01450-t002:** Participants’ problems managed in general practice encounters (3670 encounters from 432 adolescents).

Problems Managed	Rate per 100 Encounters
Mental and behavioural problems	14.4
Behavioural problem	6.7
Depressive disorder	2.0
Attention deficit hyperactivity disorder	1.1
Sleep disturbance	1.1
Anxiety disorder (or state)	0.7
Physical problems	
Skin problems	**29.4**
Cellulitis	2.8
Wart	2.2
Acne	2.2
Rash	2.1
Infected finger/toe	3.9
Naevus/mole	1.9
Boil/carbuncle	1.7
Dermatophytosis	1.6
Dermatitis, contact/allergic	1.4
Laceration/cut	1.4
Musculoskeletal problems	**13.8**
Pain in joint	2.9
Fracture	1.5
Knee symptom/complaint	1.4
Back pain	1.4
Acquired deformity of spine	1.2
Pain in limb	1.1
Sprain/strain/dislocation	1.0
Digestive problems	**10.5**
Constipation	2.5
Abdominal pain	2.5
Gastroesophageal reflux	0.6
Neurological problems	**7.2**
Epilepsy	5.6
Pregnancy, childbearing, and family planning	**6.8**
Contraceptive management	5.3
Management of pregnancy or suspected pregnancy	1.0
Respiratory problems ^^^	**6.3**
Asthma	2.8
Acute bronchitis/bronchiolitis	1.9
General and unspecified problems ^^^	**5.8**
Effect prosthetic device	2.1
Chest pain not otherwise specified	0.7
Endocrine, metabolic, and nutritional problems	**5.3**
Obesity and overweight	1.4
Hypothyroidism	1.2
Iron deficiency	1.0
Eye problems	**4.0**
Conjunctivitis	1.8
Blepharitis/stye/chalazion	1.0
Female genital problems	**3.4**
Menstruation excessive	0.9
Menstrual pain	0.7
Urological problems	**3.2**
Urinary tract infection	1.0
Dysuria/painful urination	0.5
Cardiovascular problems	**1.4**
Problems in blood, blood-forming organs, and immune mechanism	**1.1**
Ear Problems	**1.1**
Male genital problems	**0.8**

^^^ Upper respiratory tract infections and some unspecified clinical tasks were not coded.

**Table 3 children-10-01450-t003:** Prevalence of chronic conditions in participants at 16 years of age (N = 415).

	Overall N (%)	Male n (%)	Female n (%)	Risk Difference % (95% CI)
Total	415	223	192	
Any chronic mental disorder or physical condition	347 (83.6)	188 (84.3)	159 (82.8)	1.5 (−5.7, 8.7)
Any chronic mental disorder	169 (40.7)	108 (48.4)	61 (31.8)	16.7 (7.6, 25.7) **
Autism spectrum disorder	77 (18.6)	58 (26.0)	19 (9.9)	16.1 (9.1, 23.1) **
Attention deficit hyperactivity disorder	61 (14.7)	41 (18.4)	20 (10.4)	8.0 (1.1, 14.8) *
Psychiatric diseases ^&^	55 (13.3)	25 (11.2)	30 (15.6)	−4.4 (−10.9, 2.0)
Anxiety disorders	16 (3.9)	6 (2.7)	10 (5.2)	−2.5 (−6.3, 1.3)
Depressive disorders	15 (3.6)	7 (3.1)	8 (4.2)	−1.0 (−4.9, 2.9)
Any chronic physical condition	290 (69.9)	150 (67.3)	140 (72.9)	−5.7 (−14.6, 3.3)
Neurological conditions	119 (28.7)	64 (28.7)	55 (28.7)	0.1 (−8.8, 8.9)
Active epilepsy	61 (14.7)	35 (15.7)	26 (13.5)	2.2 (−4.9, 9.2)
Cerebral palsy	33 (8.0)	17 (7.6)	16 (8.3)	−0.7 (−5.9, 4.4)
Sleep apnea	11 (2.7)	5 (2.2)	6 (3.1)	−0.9 (−4.0, 2.3)
Congenital malformation or deformation	95 (22.9)	53 (23.8)	42 (21.9)	1.9 (−5.8, 9.5)
Congenital heart defects	31 (7.5)	14 (6.3)	17 (8.9)	−2.6 (−7.7, 2.5)
Respiratory conditions (excluding allergic rhinitis)	79 (19.0)	43 (19.3)	36 (18.8)	0.5 (−6.8, 7.8)
Asthma	75 (18.1)	41 (18.4)	34 (17.7)	0.7 (−6.4, 7.8)
Endocrine and metabolic conditions	63 (15.2)	31 (13.9)	32 (16.7)	−2.8 (−9.8, 4.3)
Hypothyroidism ^#^	17 (4.1)	11 (4.9)	6 (3.1)	1.8 (−2.0, 5.6)
Diabetes	3 (0.7)	2 (0.9)	1 (0.5)	0.4 (−1.2, 2.0)
Visual conditions	56 (13.5)	32 (14.4)	24 (12.5)	1.8 (−4.8, 8.5)
Visual disturbance or impairment	46 (11.1)	27 (12.1)	19 (9.9)	2.2 (−3.8, 8.2)
Strabismus ^^^	25 (6.0)	13 (5.8)	12 (6.3)	−0.4 (−5.0, 4.2)
Musculoskeletal conditions	54 (13.0)	33 (14.8)	21 (10.9)	3.9 (−2.8, 10.5)
Scoliosis	24 (5.8)	16 (7.2)	8 (4.2)	3.0 (−1.4, 7.4)
Skin conditions	39 (9.4)	21 (9.4)	18 (9.4)	0.0 (−5.6, 5.7)
Melanocytic nevus	17 (4.1)	9 (4.0)	8 (4.2)	−0.1 (−3.5, 3.3)
Atopic dermatitis	16 (3.9)	7 (3.1)	9 (4.7)	−1.5 (−5.3, 2.2)
Cardiovascular conditions ^§^	38 (9.2)	19 (8.5)	19 (9.9)	−1.4 (−6.9, 4.2)
Valvular heart diseases	9 (2.2)	4 (1.8)	5 (2.6)	−0.8 (−3.7, 2.0)
Other conditions				
Hearing loss	20 (4.8)	8 (3.6)	12 (6.3)	−2.7 (−6.9, 1.6)
Gastroesophageal reflux	18 (4.3)	9 (4.0)	9 (4.7)	−0.7 (−4.5, 3.2)

CI: confidence interval; * *p* < 0.05; ** *p* < 0.001; ^&^ Psychiatric diseases are mental disorders with exclusion of developmental disorders (including autism spectrum disorder, attention deficit hyperactivity disorder, Tourette Syndrome, specific speech and language disorders, specific scholastic skills disorders, specific developmental disorders of motor function). Most of developmental disorders cases were autism spectrum disorder and attention deficit hyperactivity disorder; ^#^ No case of iodine-deficiency hypothyroidism was recorded; ^^^ All strabismus cases were non-paralytic strabismus; ^§^ Cardiovascular conditions include congenital heart defects.

**Table 4 children-10-01450-t004:** Incidence of chronic conditions (N participant = 432).

Chronic Medical Conditions or Mental Disorders	Incidence per 1000 Person-Years (95% CI) ^&^
Mental disorders	
Depressive disorders	7.7 (4.5–13.3)
Anxiety disorders	6.5 (3.6–11.7)
Autism spectrum disorder	0.7 (0.1–4.9)
Attention deficit hyperactivity disorder	0.7 (0.1–4.7)
Physical disorders	
Gastroesophageal reflux	9.4 (5.8–15.4)
Atopic dermatitis	8.8 (5.3–14.5)
Amenorrhea	5.1 (2.7–9.8)
Asthma	4.8 (2.3–10.1)
Hearing loss	4.8 (2.4–9.5)
Hypothyroidism ^#^	4.7 (2.3–9.3)
Sleep apnea	4.6 (2.3–9.2)
Active epilepsy	3.5 (1.5–8.4)
Seborrhoeic dermatitis	2.9 (1.2–6.9)
Visual impairment	2.5 (1.0–6.8)
Abnormal involuntary movements	1.7 (0.6–5.3)
Strabismus	1.2 (0.3–4.8)

CI: confidence interval; ^#^ No case of iodine-deficiency hypothyroidism was recorded; ^&^ Total person-years at risk for each medical condition or mental disorder ranged from 1430 to 1767 person-years.

## Data Availability

The data presented in this study are available on request from the corresponding author. The data are not publicly available due to patient privacy requirements related to the clinical data.

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
