# Peer review of "Reasons for Encounters and Comorbidities in Adolescents with Intellectual Disability in General Practice: A Retrospective Analysis of Data from the Ask Study"

_children, 2023, doi:10.3390/children10091450_

Round 1

Reviewer 1 Report

This paper describes problems managed during primary care presentations, and the prevalence and incidence of chronic medical conditions in Australian adolescents with intellectual disability .A good comparison of the data obtained in this study is made with the literature on general population and comparison with other literature on people with intellectual disability. However, some aspects need clarification:

    2.2. Data collection 103- 104 lines “Data from primary care consultations were collected from GP notes by research staff between 10 September 2009 and 10 June 2010”. The data are from a long time ago, are there no updated data?

      Skin problems (29.4 per 100 encounters) were the most common conditions managed within primary care. Please provide justification or further explanation.

    303-321 lines: This paragraph could be summarized as it includes a lot of information regarding plans in Australia that may not be of interest to the reader. Highlight only the implications of general interest.

− Has the degree of intellectual disability of study participants been measured? This can be highly variable and measured with various tests. It is a descriptive study, but as a limitation of the study, I would include that the degree of intellectual disability has not been meausured o that has not been related to described chronic physical conditions and mental disorders

Reviewer 2 Report

The authors explore the reason for encounter (RFE), and the prevalence and incidence of co-occurring of medical conditions in adolescents with intellectual disability in general practice, by reanalyzing data from “Ask Study”.  This paper will be well received by a broader audience including general practitioners, pediatricians, psychiatrists, neurologists, etc.  I would use standard terms instead of general terms in the paper so that it would become comprehensive and gain more hits in searches in electronic databases. My comments are just minor fixes, which I believe will bring the manuscript to the next level.

Comments  

Title and Abstract

The term “Presentations” in the title reflects only the “reason for encounter”, but not the co-occurring conditions” explored in the study. And, I would suggest using standard terms such as “reason for encounter” (RFE), or “presenting problem” instead of “Problems managed”.

An alternative title to consider; is “Reasons for Encounter and comorbidities in adolescents with intellectual disability in general practice; a retrospective analysis of data from the Ask Study”

Please consult the following paper.

Weise J, Pollack A, Britt H, Trollor JN. Primary health care for people with an intellectual disability: an exploration of demographic characteristics and reasons for encounters from the BEACH programme. J Intellect Disabil Res. 2016 Nov;60(11):1119-1127. doi: 10.1111/jir.12301. Epub 2016 Jun 9. PMID: 27278719.

“The study explored three main components; 1) reason for encounter in primary care 2) prevalence of comorbidities at age of 16 years, and 3) incidence of comorbidities through adolescence.”

I feel it is worth mentioning in the abstract, or the title that this study is an “A retrospective analysis of data from the Ask Study”

“Ask study” is Ask an acronym? If so, should be defined at the first mention.

Indicate the trial registration number of the Ask Study for quick reference (ClinicalTrials.gov Identifier: NCT00519311)

Have you used any diagnostic criteria/ diagnostic tool for Intellectual disability or is it just a professional diagnosis? If so, mention it in the inclusion criteria.

Results

readers would like to know the rates of underlying genetic conditions other than Down syndrome. Is it possible to reanalyze and mention a breakdown of the rates of other known genetic conditions? This will give further characterize your population and help to understand their primary healthcare encounters and comorbidities.
